# Deep Learning Based Automatic Malaria Parasite Detection from Blood Smear and Its Smartphone Based Application

**DOI:** 10.3390/diagnostics10050329

**Published:** 2020-05-20

**Authors:** K. M. Faizullah Fuhad, Jannat Ferdousey Tuba, Md. Rabiul Ali Sarker, Sifat Momen, Nabeel Mohammed, Tanzilur Rahman

**Affiliations:** Department of Electrical & Computer Engineering, North South University, Dhaka 1229, Bangladesh; faizullah.fuad@northsouth.edu (K.M.F.F.); jannat.ferdousey@northsouth.edu (J.F.T.); sarker.rabiul@northsouth.edu (M.R.A.S.); sifat.momen@northsouth.edu (S.M.); nabeel.mohammed@northsouth.edu (N.M.)

**Keywords:** Plasmodium parasites, microscopic, blood smear, data augmentation, CNN, knowledge distillation, Autoencoder, inference performance, floating point operations, deep learning

## Abstract

Malaria is a life-threatening disease that is spread by the Plasmodium parasites. It is detected by trained microscopists who analyze microscopic blood smear images. Modern deep learning techniques may be used to do this analysis automatically. The need for the trained personnel can be greatly reduced with the development of an automatic accurate and efficient model. In this article, we propose an entirely automated Convolutional Neural Network (CNN) based model for the diagnosis of malaria from the microscopic blood smear images. A variety of techniques including knowledge distillation, data augmentation, Autoencoder, feature extraction by a CNN model and classified by Support Vector Machine (SVM) or K-Nearest Neighbors (KNN) are performed under three training procedures named *general training*, *distillation training* and *autoencoder training* to optimize and improve the model accuracy and inference performance. Our deep learning-based model can detect malarial parasites from microscopic images with an accuracy of 99.23% while requiring just over 4600 floating point operations. For practical validation of model efficiency, we have deployed the miniaturized model in different mobile phones and a server-backed web application. Data gathered from these environments show that the model can be used to perform inference under 1 s per sample in both offline (mobile only) and online (web application) mode, thus engendering confidence that such models may be deployed for efficient practical inferential systems.

## 1. Introduction

Malaria, a life life-threatening disease caused by Plasmodium parasites, is still a severe health concern in large parts of the world especially the third world countries. Almost 219 million cases of malaria, across 87 countries worldwide, were reported by World Health Organization (WHO) in the year of 2017 [1]. WHO designated South-East Asia, Eastern Mediterranean, Western Pacific, and the Americas as high risk zone. Malaria is curable and can be prevented if proper initiatives and approaches are taken effectively, which majorly relies on early diagnosis of malarial parasites [2].

Many techniques have been reported, in the literature, which detects malarial parasites in a patient such as clinical diagnosis [3,4], microscopic diagnosis [5,6,7,8,9,10], rapid diagnostic test (RDT) [11,12,13] and polymerase chain reaction (PCR) [14,15,16]. Conventional diagnostic methods like clinical diagnosis and PCR are performed in laboratory settings, and their efficiency and accuracy largely depend on the level of available human expertise [17]. Such expertise is inadequately present in unreached remote areas where malaria can be predominant.

RDT and microscopic diagnosis are two of the most impactful malaria diagnosis methods which make a very large contribution to malaria control today [13]. RDT is effective diagnosis tool as it does not require any trained professional or microscope and can provide diagnosis within 15 min [18]. However, according to WHO [19] and others [20,21] RDT has few shortcomings that includes lack of sensitivity, inability to quantify parasite density and differentiate among *P. vivax*, *P. ovale* and *P. malariae*, higher cost compared to light microscope, and susceptibility to damage by heat and humidity. Microscopic systems do not suffer from these shortcomings and is considered to be effective for malaria parasite detection [13,22], however this technique requires the presence of a trained microscopists [23]. Automatic microscopic malaria parasite detection, which involves the acquisition of the microscopic blood smear image (for example by smartphone as demonstrated in [24,25]), segmentation of the cells and classification of the infected cells, can be an effective diagnostic tool [26]. It is to be noted that successful segmentation of blood cells and identification of malaria parasites can be used in conjunction to perform counting. Cell segmentation is a well researched area and good performance has already been reported in multiple studies [27,28,29,30]. Previous works that concentrated on the classification of infected cells involve tools and techniques from image processing [31,32,33], computer vision [34,35,36] and machine learning [37,38,39]. However, solutions for classification of infected cells, which are both accurate and computationally efficient, have not been studied to the best of our knowledge. As an example, [40] reports an accuracy of 99.52% but the proposed model uses over 19.6 billion floating point operations (flops). This precludes the model from being useful in power constrained device.

In this study we trained multiple accurate and computationally efficient models for malaria parasite detection in single cells using a publicly available malaria dataset [24]. Our contributions reported in this article are mainly three-folds:While doing experiments on the malaria dataset, we found that certain samples were mislabeled. These mislabeled samples were corrected and the corrections can be found in [41] for future research.Unlike previous works carried out for malaria parasite detection, our trained models are not only highly accurate (99.23%) but also the order of magnitude is more computationally efficient (4600 flops only) compared to previously published work [40].For better understand the performance of the model in low resources settings, it was deployed for inference in multiple mobile devices as well as a web application. We find that in such situation the model can be used to find accurate per cell classification prediction within 1 s.

It is conceivable that these contributions can play a significant role towards building a fully automated system for malaria parasite detection in the future.

## 2. Relevant Work

Malaria being a life threatening disease has caused deep research interests among the scientists all over the world. Earlier, malaria was mostly diagnosed in the laboratory setting requiring a great deal of human expertise. Automatic systems such as those relying on machine learning techniques were initially studied to overcome this problem. Techniques reported in this domain of study mostly considered the hand-crafted features in decision making. For example, [42,43] relied on morphological factors for feature extraction and applied SVM and Principal Component Analysis (PCA) [44] for the classification purpose. However, the accuracy achieved through these kinds of model is low compared to the more recently studied deep learning based techniques.

For automatic detection of malaria pathogens from the microscopic images, Convolutional Neural Network (CNN) [45] received much attention from the researchers in recent times. Dong et al. for example evaluated the performances of three popular Convolutional Neural Network [46], namely LeNet-5 [47], AlexNet [48] and GoogLeNet [49]. In addition, they trained an SVM classifier for comparison purposes and concluded that CNN is advantageous over SVM in terms of the capacity of learning image features automatically. To find the optimal layer of a pre-trained model to extract features from underlying malaria parasite data, Rajaraman et al. evaluated the performances of AlexNet, VGG-16 [50], ResNet-50 [51], Xception [52], DenseNet-121 [53] along with their custom-built model [24]. Liang et al. reported 97.37% accuracy in detection with the help of their 16-layered CNN model and claimed that their model is superior to transfer learning models [54]. Hung et al. pre-trained a model on Imagenet [55] but fine-tuned it on own data for detecting malaria parasite [35].

Most of the reported studies in general focused on improving the accuracy using traditional machine learning or deep learning based techniques. A few notable exceptions are the works of Quinn et al. [25] and Rosado et al. [56] who also tackled the problem of computational efficiency of their models. However, both studies reported a notable drop in model accuracy in pursuit of computation efficiency. Rosado et al. also explored the use of smartphones to detect malaria parasites and white blood cells. They reported sensitivity and specificity of 80.5% and 93.8% for trophozoite detection and 98.2% and 72.1% respectively for white blood cells using Support Vector Machine (SVM) [57]. Using the same classifier, they achieved a sensitivity and specificity of 98.2% and 72.1% respectively for white blood cells. The use of deep learning technique, particularly those that leverage transfer learning [31,40,58] yield superior results when considering classification metrics. Unfortunately, the models proposed in those studies are also quite expensive in terms of the required computational resources. Also notable is the fact that while Rosado et al. showed that their model can be deployed on mobile phones, their demonstration did not include low cost mobile phones commonly used in the poorer regions of the world.

In contrast, in this work, we propose several deep learning models which achieve classification performance comparable to the previously reported highly accurate deep learning based solutions. In addition, our models are efficient in terms of required computational resources and have been demonstrated to work efficiently on smart mobile devices, including that are available at very low cost.

## 3. Methodology

In order to conduct a series of experiments, publicly available malaria dataset was used. Data collection and data preprocessing techniques are discussed in the following subsequent subsections. Out of the series of experiments, we choose our best model in terms of both performances and effectiveness thereby, which is discussed in proposed model architecture subsections. Experimental details and experimental settings are discussed in training details subsections. Training of the models is discussed under three training procedure which are *general training* procedure, *distillation training* procedure and *autoencoder training* procedure, details are provided in the designated subsections.

### 3.1. Data-Set

Malaria dataset contains 27,558 cell images classified into two groups called parasitized and uninfected cells, where each cell contains an equal number of instances. Data was taken from 150 P. falciparum and 50 healthy patients and it was photographed at Chittagong Medical College Hospital, Bangladesh using a smartphone by placing it on the conventional light microscope [26]. Manual annotation was performed later by an expert slide reader at the Mahidol-Oxford Tropical Medicine Research Unit. In this data, parasitized samples mean that there is the presence of Plasmodium, whereas the uninfected samples refer to the absence of Plasmodium but there may be presence of other objects like staining artefacts/impurities.

While studying the dataset, some of the labelled data raised suspicion of whether they were correctly labelled. Some of the data seems like uninfected but labelled as parasitized, where some parasitized images are labelled as uninfected. To confirm this rising issue we consulted with an expert. The expert confirmed that some of the data are genuinely mislabeled which was later manually annotated as per the presence and absence of malarial parasites. While annotating, suspicious, and falsely labelled data was put aside, which resulted in the reduction of data from 27,558 to 26,161. After removing 647 falsely labeled and suspicious parasitized data, the amount of current parasitized data stands 13,132. In this article, correct parasitized data is considered as true parasitized, and suspicious data is considered as false parasitized. For uninfected malaria data, 750 suspicious images was found, which was named as false uninfected. After keeping away those data, total true uninfected data stands 13,029. Some data from the dataset are depicted in Figure 1.

### 3.2. Data Preprocessing

In supervised learning, the behaviour and performances of the model entirely depends on the data that is fed. Therefore, data preprocessing plays a vital role towards conducting experiments. Considering that, in this work manually corrected images were resized as per the model input, and image patches were rescaled to map the features within 0 to 1 range which led to getting faster convergence. Data augmentation as per Table 1 was also applied to training data preserving the semantic meaning, which helps to improve the model performances. Figure 2 depicts resizing augmented images.

### 3.3. Proposed Model Architecture

To serve the purpose of detecting malaria parasite from blood smear (exactly the similar kind of blood smear collected by [24]), in this article, an autoencoder-based architecture is proposed, which is shown in Figure 3. Autoencoder [59] is a specific type of artificial neural network which compresses input data into lower-dimensional latent space representation and finally reconstruct output from this representation shown in Equations (1) and (2).
(1)$$\bar{X_{i}}=Decoder\left(Encoder\left(X_{i}\right)\right)$$
(2)$$L_{\theta }=C_{\theta }\left(\bar{X_{i}},\;X_{i}\right)$$

Here, Mean Square Error (MSE) loss function *C* calculates the loss between *i* number of the original and reconstructed image.

The primary purpose of an Autoencoder is dimensional reduction [60]. However, in this task, it is used as a classifier, inspired by [61,62]. Autoencoder is composed of two main components: Encoder and Decoder. For the classification task, the decoder is replaced by the fully connected layer which allowed Autoencoder to classify the expected classes. The complete process will be discussed in the subsequent section.

**Encoder:** Encoder compresses the input to latent space representation with the least possible distortion. For *X_n_* = *x*_1_, *x*_2_, *x*_3_, …, *x_n_* set of input images belongs to training set, encoder compress it to *K_n_* = *k*_1_, *k*_2_, *k*_3_, …, *k_n_* where *K_n_* is the set of latent representation of *X_n_*.
(3)$$k_{1},k_{2},k_{3}\ldots \ldots k_{n}=Encoder\left(x_{1},\;x_{2},\;x_{3}\ldots \ldots x_{n}\right)$$
(4)$$K_{n}=Encoder\;\left(X_{n}\right)$$

The proposed encoder is composed of three Convolutional layers where a Max-pooling layer follows each layer. For performing the convolutional operation in each layer, the kernel size is defined as (3 × 3) with the same padding and 1-pixel stride. The kernel number for the first convolutional layer was set 16 where second and third layers were respectively 8 and 4. ReLU activation function, shown in Equation (5), was applied in encoder’s hidden units to introduce non-linearity to the neuron’s output.
(5)$$S=\max\left(0,M\right)$$

Here, *S* is the output after applying non linearity on matrix *M*.

To sample down the features map, max-pooling layer shown in Equation-6 is applied with window size 2 × 2 and strides 1.
(6)$$Z=max_{i,j=1}^{h,w}\left(M_{i,j}\right)$$

*Z* is the output matrix containing maximum value of each patch from input matrix *M*.

**Decoder:** Decoder reconstructs the image *R_n_* = *r*_1_, *r*_2_, *r*_3_, …, *r_n_* from the latent representation *K_n_* = *k*_1_, *k*_2_, *k*_3_, …, *k_n_* is shown in Equations (7) and (8).
(7)$$r_{1}\;,r_{2}\;,r_{3}\ldots \ldots r_{n}=Decoder\;\left(k_{1}\;,k_{2}\;,k_{3}\ldots \ldots k_{n}\right)$$
(8)$$R_{n}=Decoder\;\left(K_{n}\right)$$

The decoder consists of 4 Deconvolutional layers and three Up-sampling layers. The kernel size for all the Deconvolutional layers is 3 × 3 with strides size 1 having the same padding and number of kernels were defined as 4, 8, 16 and 3 respectively. Deconvolutional layer is the opposite of convolutional layer and unlike Convolutional layer, instead of mapping 3 × 3 features into 1 pixel, Deconvolutional layers map 1 pixel to 3 × 3 features vector. ReLU activation function was applied to the hidden units to introduce non-linearity. Up-sampling with window size 2 × 2 was applied to get closer input image to reconstruct it from the latent representation.

While testing, decoder is replaced by a flatten and two fully connected layers to serve our purpose of detecting malaria parasite from a blood smear. Hence, the final model is composed of 9 layers among which three are Convolutional layers, three are Max-pooling layer, one Flatten layer and two fully connected layers. Finally, fully connected layer is followed by a softmax layer to distribute predicted probabilities over two classes according to the objective of this task. The outline of our proposed model architecture is given in Figure 3.

### 3.4. Training Details

To find the optimum and efficient model, different approaches was investigated. Among the series of experiments, feature extractions and classification using Support Vector Machine and K-Nearest Neighbors (KNN) along with augmentation and without augmentation is discussed under General training procedure. In Distillation training procedure, model pruning and knowledge transformation from teacher to student model is discussed. The third procedure is the Autoencoder training procedure where training and testing procedure of an Autoencoder based model is discussed. For general training and knowledge distillation training, a custom 8-layered Convolutional Neural Network model, shown in Figure 4, was used as a feature extractor. In all the training procedure, total data was divided into training, testing and validation group as per the ration 80:10:10. All experiments was performed on machine Ubuntu 16.04, system with Intel@ Core i5-6500 CPU @ 3.20 GHz processor, 8 GB RAM, 1 TB HDD, Python @3.6.7, Keras @2.2.4 having TensorFlow @1.13.0 backend.

#### 3.4.1. General Training

In order to optimize and to obtain better performances, data augmentation, without augmentation, classification using SVM and KNN was performed. For the augmentation experiment, augmentation as per the Table 1 was applied first. While performing 64 × 64 augmentation experiment, original image was resized to (64, 64) before applying augmentation where augmentation was performed after resizing original image into (32, 32) for the experiment of 32 × 32 augmentation. In both experiments, data was fed into the model according to batch size 32. For the without augmentation training, plain data was fed into the model according to batch size 64 after resizing the original image to (64, 64) or (32, 32) as per the experiment requirement. Weights and biases of all trainable parameters were randomly initialized at the beginning of training. To optimize the weights and biases, Stochastic Gradient Descent (SGD) [63] optimizer was used while performing the forward propagation. The learning rate was set to 0.001 with a momentum of 0.9. While performing back-propagation to calculate the error and update the weights and biases, cross entropy, shown in Equation (9), was used as a loss function. Predicted result and combined validation loss and accuracy of augmentation and without augmentation experiment for both image sizes are depicted in Figure 5; Figure 6, respectively.
(9)$$Loss=-\left(y\;log\left(p\right)+\left(1-y\right)\log\left(1-p\right)\right)$$

In the experiment of CNN-SVM and CNN-KNN, a CNN model was used to extract the features. Later these features were classified through SVM or KNN as per the objective of our experiment. The outline of this architecture is depicted in Figure 7.

Support vector machine (SVM) and K-Nearest Neighbor (KNN) are the core machine learning algorithms that were used as a classifier by implicitly mapping their inputs into high-dimensional feature spaces [57,64]. Earlier work [65,66,67] reported the effectiveness of SVM or KNN classifier on the Convolutional neural network’s extracted features instead of using the softmax layer. Being motivated with this work, we trained SVM and KNN models after extracting features from a CNN model (depicted in Figure 4) as shown in Equations (10) and (11).
(10)$$Label_{\propto }^{pred}=S\left(C^{l-1}\left(X_{\propto }\right)\right)$$
(11)$$Label_{\propto }^{pred}=K\left(C^{l-1}\left(X_{\propto }\right)\right)$$

Here, *X* is the images of training set, *C* is the feature extractor model, *l* is the number of layer and *S*, *K* is respectively SVM and KNN classifier.

While training CNN-SVM or CNN-KNN model, to get better performance, hyperparameters were fine-tuned. In CNN-SVM model 98.93% accuracy was achieved by using ‘RBF’ kernel and setting gamma and regularization parameter value to 0.001 and 10 respectively. The best accuracy of CNN-KNN model was 99.12% when 3 was defined as neighbor’s number. Model performances under general training procedure are shown in Table 2.

#### 3.4.2. Distillation Training

As a part of the experiment, model pruning method like knowledge distillation was also explored. Knowledge distillation is a model compression method shown in Equation (12) where knowledge is transferred from a larger model (teacher model) to a smaller model (student model). In this method, a smaller model is trained to learn the exact behavior of a complex and bigger model. This method is proposed by Bucila [68] and generalized by Hinton [69].
(12)$$Z_{i}=\frac{\exp\left(\frac{k_{i}}{T}\right)}{\sum_{j}\exp\left(\frac{k_{i}}{T}\right)}$$

In our experiment, a pretrained 146 layers custom ResNet model [70] was taken as a teacher model, whereas a custom 8 layers CNN model (showed in Figure 4) acted as a student model. In order to mimic the teacher model, initially using the pretrained teacher model, the softmax value of training images was recorded and also the training label was predicted. Later, the softmax value of training images was swapped if the teacher model mis-predicts the training label. This corrected softmax value was used as a training label while training student model, shown in Equations (13) and (14).
(13)$$\gamma_{i}^{\tau }=T_{m}\left(X_{i}\right)$$
(14)$$\gamma_{i}^{S_{train}}=\left\{\begin{array}{ll} \gamma_{i}^{\tau }, & if\;argmax\;\left(\gamma_{i}^{\tau }\right)==argmax\;\left(\gamma_{i}^{\rho }\right)\\ \gamma_{i}^{\rho }, & Otherwise \end{array}\right.$$

Here, $\gamma_{i}^{S_{train}}$ is training label of student model

$\gamma_{i}^{\tau }$ is the softmax prediction of teacher model

$\gamma_{i}^{\rho }$ is the actual label

$T_{m}$ is the Teacher model

Stochastic Gradient Descent (SGD) was used as an optimizer with initial learning rate of 0.01 and momentum of 0.5 while training the student model. To update the weight, a decay of 1e-6 was applied using Keras learning rate decay function. To calculate the loss and update the weights and biases of all trainable parameters Cross Entropy Loss function shown in Equation (9) was applied. Combine validation accuracy and loss is shown in Figure 8, and the predicted result of this model are shown in (Appendix A).

After training model performance are depicted in Table 3.

#### 3.4.3. Autoencoder Training

Finally, autoencoder based architecture was explored to increase the model’s performance. It is found that the autoencoder based experiments outperforms other experiments. Autoencoder is composed of two main components which are Encoder and Decoder discussed in the proposed model subsection. The main aim of an Autoencoder is dimensionality reduction, but to detect malaria parasite, it is used as a classifier. This classifier can classify infected and parasitized images from a microscopic blood smear. To train the classifier, at first, full Autoencoder network was trained with the augmented training images where augmentation was performed as per Table 1. Mean Square Error (MSE) loss function shown in Equation (15) was used while training to calculate the losses. To minimize the loss and update the weights RMSprop [71] optimizer was used with 0.001 learning rate. Figure 9 depicts the few reconstructed images after training Autoencoder 500 epochs.
(15)$$Loss_{MSE}=\frac{1}{k}\sum_{j=1}^{k}{\left(X_{i}-{\bar{X}}_{i}\right)}^{2}$$

After training Autoencoder network, Decoder was replaced by a fully connected layer where a softmax layer follows the fully connected layer in order to distribute predicted probabilities into uninfected and parasitized class. This modified network was trained for 300 epochs after freezing encoder layers which helps fully connected layers to learn the image features without changing encoder’s weights and biases. Cross Entropy loss functions was used to calculate the losses. SGD optimizer with learning rate 0.001 and momentum 0.9 was used to optimize and update the weights and biases. Figure 10 depicts the training behavior of this modified model.

Further, Autoencoder model was trained after unfreezing the encoder layer, which led to achieving better performance by the model. While training, the hyper-parameter was set as like before. The experiment was done for both 32 × 32 and 28 × 28 images. The final behavior of the model in both experiments is depicted in Figure 11 and Table 4.

## 4. Result Analysis

To develop an efficient and highly accurate model for the detection of the malaria parasite from segmented cell images, a series of experiments involving both machine learning and deep learning techniques were performed. The models were evaluated using different performance metrics such as Test Accuracy (Test Acc), F1 Score, Precision (Precs.), Sensitivity (Sens.) and Specificity (Spec.). It is important to note that the size of the model was also considered as an important factor along with performance to ensure the viability of the models in lower-cost smartphones. The performance of our models explained by the above-mentioned evaluation metrics can be found in Table 5.

Table 5 depicts the performance of the models on the test data as obtained from the experiments. In terms of size and model performance, models trained using distillation, and autoencoder is compatible to serve the purpose of being computationally efficient while retaining high classification accuracy. As the autoencoder trained model is smaller in size (only 73.70 KB) and outperforms models obtained from distillation and other techniques, these models may be considered for practical validation on smartphones. The two autoencoder based models differ in their input image resolution. One uses 28 × 28 images and the other accepts 32 × 32 images. Figure 12 and (Appendix A) shows the prediction of autoencoder model experimented on 32 × 32 images and the image resolution of the two respective autoencoder based models. Although the model that accepts 28 × 28 images has slightly higher accuracy, we preferred to use the model which uses the slightly higher resolution image for deployment in mobile phones and the web-application.

Autoencoder model experimented on 32 × 32 falsely predicts 23 images while testing on 3000 test images. The confusion matrix is placed in Figure 13 to show the model performances on test data.

According to Figure 13, it is clear that among the 23 falsely classified images, 16 are false positive, and 7 are false negative. Some of the misclassified images are shown in (Appendix A). Studying upon all those images, we found that, some of the images are hard to classify even for the human eyes. So, with the special consideration of this issue, we can say that autoencoder based model performs well on malaria data.

In addition, the generalizability of the model was established by evaluating its performance on images from a separate publicly available dataset [72]. The comparable performance achieved with this dataset further confirm that the proposed model is not localized or biased to the dataset on which it was trained. Experimental details and results on this new dataset can be found in the Appendix B.

## 5. Model Deployment

To demonstrate the robustness and compatibility of the developed model, both web-based and mobile-based applications were developed. This development clearly shows that this model can be beneficial for the automatic diagnosis of Malaria in a restricted resource environment.

### 5.1. Mobile Based Application

Mobile phone has become ubiquitous these days and is an essential part of our day to day life activities. It is not only used for communication but also uses in many other purposes. Mass people use this powerful device even in a developing country due to being available at a reasonable cost. Keeping those in mind, the developed model was integrated with a smartphone application to demonstrate that Smartphone’s based automatic detection of the malaria parasite is quite possible. The model can work independently in the mobile app without needing internet connection and can help an individual without any technical expertise to detect malaria parasite from the blood smear. Figure 14 depicts the pictorial diagram of a mobile-based solution.

To deploy a model into the smartphone, the model needs to be converted into TensorFlow Lite model as it is the best solution to execute the trained model accurately in limited memory and computational power. Along with this, TensorFlow Lite converter provides options that allow to reduce file size and increase the speed of execution, with some trade-offs. Considering all those, for making mobile-based application at first, trained model was converted into TensorFlow Lite. After that, an interpreter was initialized for loading the model. An interpreter is a program that converts programs written in a high-level language into machine code understood by the computer. The model was then initialized after it was appropriately loaded. To process image, the mobile application provides the option for capturing image from the storage. After selecting the image from the gallery or taking a picture from the camera resizing of the image into 32 × 32 is performed as per model input. Later, it is passed to the CNN model as an input for invoking the interpreter to get the result. Then the model can run and analyze the image. After the analysis, the result would be given by the model in binary, which would represent whether the result is parasitized or uninfected.

The validity of this developed model was tested on different Smartphone. Table 6 shows the detailed information of testing result on the different operating system and its resources specification.

The graphical interface of developed mobile based application is shown is Figure 15.

### 5.2. Web Based Application

A web-based application was developed to demonstrate the model compatibility on server. This application was developed using Flask. As it is a part of research work, service was kept in local machine apart from deploying in online. Figure 16 depicts the behavior of this model in the web-based application.

## 6. Discussion

In this article, an efficient and highly accurate model is developed that detects malarial parasites from microscopic blood smear. In order to assess the suitability of deploying the model in power constrained devices, we conducted a series of experiments. Out of these experiments, we found that autoencoder outperforms other approaches. The reported accuracy of the autoencoder model is 0.9923 whereas, to the best of our knowledge, the best model performance, in terms of accuracy, was reported to be 0.9952 (Shown in Table 7). Rajaraman et al. used an ensemble of VGG-19 and SqueezeNet model to achieve this performance. However, their model lacks severely in terms of efficiency which makes them unsuitable to be deployed in power constrained devices. VGG-19 alone requires 19.6 billion flops whereas our model required a total of nearly 4600 flops. It clearly shows that our model is thousands times more efficient compared to the best reported performances.

The model developed can not only diagnose malarial parasites efficiently but also maintains high accuracy. Experimented model with image 28 × 28 achieves an accuracy of 0.9951 (Showed in Table 5) which is almost equal to the reported best performance. Considering the image quality, we proposed the model experimented on image 32 × 32 but this did not compromise too much accuracy (only 0.0029). From Table 5; Table 7, it is clearly evident that our model performs significantly well in terms of Sensitivity, Specificity and Precision compared with the current state of the art performances.

To show the efficiency and compatibility of our model, we deployed our model in both web-based and smartphone-based platforms. We also tested on different resource constrained smartphone-based systems as shown in Table 6. From the table it is evident that our model has compatibility to perform inference on any smartphone that is available in the market. To perform inference, it typically takes less than a second. In existing reported works, some of the model cannot even be deployed in any portable devices whereas our model can perform inference offline in nearly all market available power constrained devices. It clearly indicates the possibility of developing portable power constrained malaria parasite diagnostic systems for undeveloped, resources restricted, unprivileged communities.

## 7. Conclusions and Future Work

This paper presents multiple classification models for malaria parasite detection which take into consideration not only classification accuracy but also aim to be computationally efficient. In the process, we conducted a series of experiments including that of general training, distillation training and autoencoder training resulting in a comparison of 10 different models. From these the best performing model achieved an accuracy of 99.5% when trained using an Autoencoder based training method on 28 × 28 images, which is comparable to the performance reported by Rajaraman et al. We find a comparable accuracy of 99.23% when we trained on 32 × 32 images. We chose to do practical evaluations using this latter model due to the slightly higher image resolution with negligible difference accuracy. It is worth mentioning that this model requires only about 4600 flops compared to over 19.6 billion flops required for the model found in previously published work. Practical validation of model efficiency was performed by deploying the model in 10 different mobile phones of varying computational capacity and a server-backed web application. Data gathered from these environments show that the model can be used to perform inference under 1 s per sample in both offline (mobile only) and online (web application) mode. These inference speeds coupled with the high classification accuracy lead us to believe that this work can play a part towards building a fully automated system for malaria parasite detection which may be useful in resource-constrained areas in the foreseeable future.

## Figures and Tables

**Figure 1 diagnostics-10-00329-f001:**
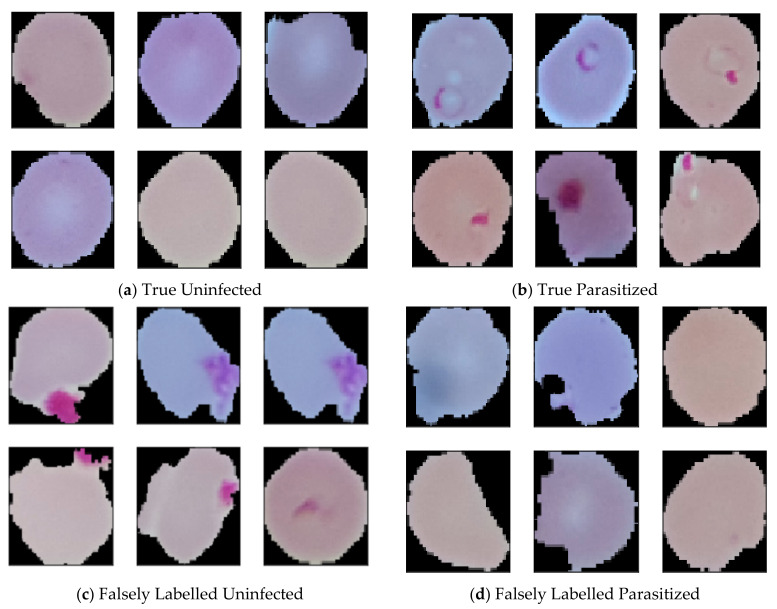
(**a**) Correctly labelled uninfected images (**b**) Correctly labelled parasitized images (**c**) Falsely labelled uninfected images and (**d**) Falsely labelled parasitized images.

**Figure 2 diagnostics-10-00329-f002:**
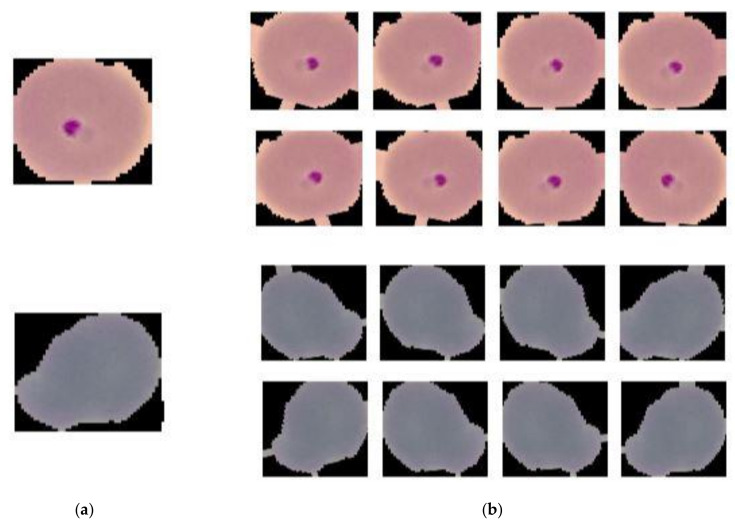
Image before resizing and performing augmentation is depicted in (**a**) and image after resizing and applying augmentation is depicted in (**b**).

**Figure 3 diagnostics-10-00329-f003:**
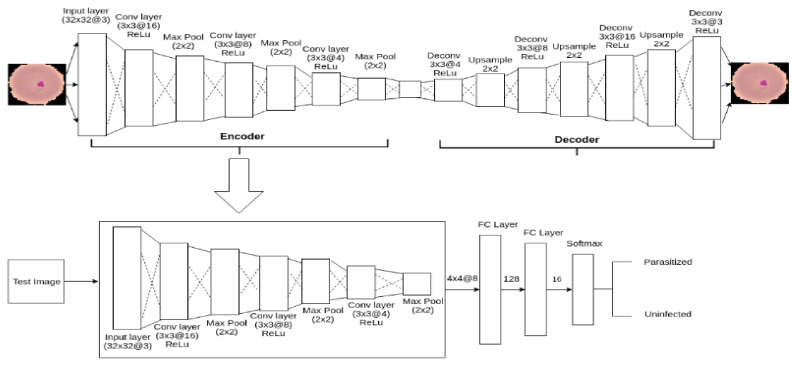
The outline of autoencoder model.

**Figure 4 diagnostics-10-00329-f004:**
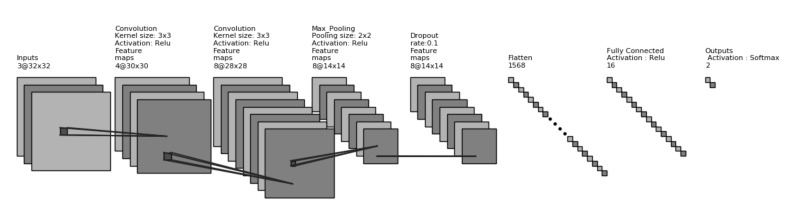
Custom 8 layers model architectures which is taken as a feature extractor for the general training procedure and knowledge distillation training procedure.

**Figure 5 diagnostics-10-00329-f005:**
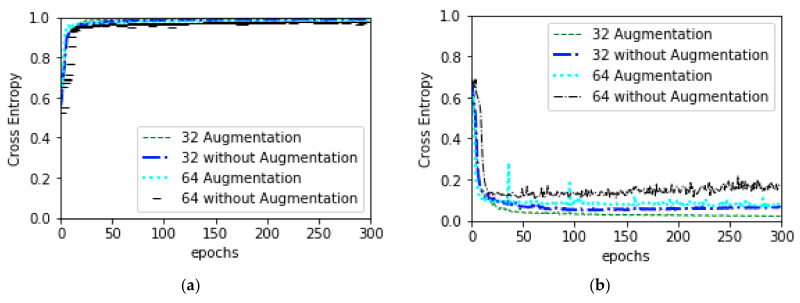
Combine validation accuracy and validation loss: (**a**) depicts combine validation accuracy and (**b**) depicts combine validation loss.

**Figure 6 diagnostics-10-00329-f006:**
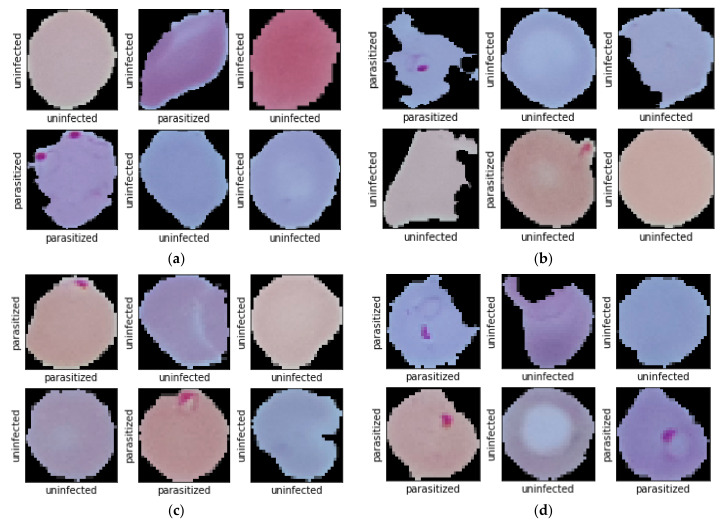
Predicted results where X-axis represents ground-truth value and Y-axis represents predictions of the model: (**a**) depicts predicted result of image 64 × 64 with Augmentation (**b**) depicts predicted result of image 64 × 64 without Augmentation (**c**) depicts predicted result of image 32 × 32 with Augmentation and (**d**) depicts predicted result of image 32 × 32 without Augmentation.

**Figure 7 diagnostics-10-00329-f007:**
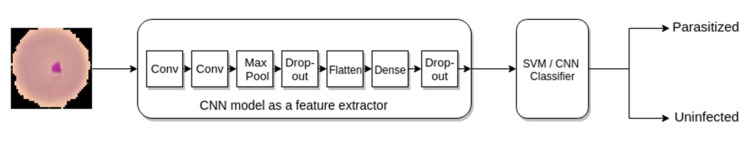
Modified CNN-KNN or CNN-SVM architecture.

**Figure 8 diagnostics-10-00329-f008:**
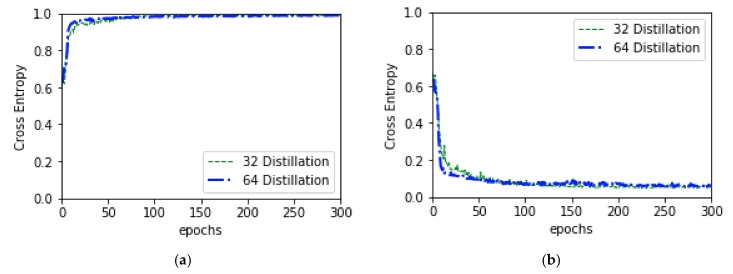
Combine validation accuracy and validation loss of knowledge distillation training: (**a**) depicts combine validation accuracy and (**b**) depicts combine validation loss.

**Figure 9 diagnostics-10-00329-f009:**
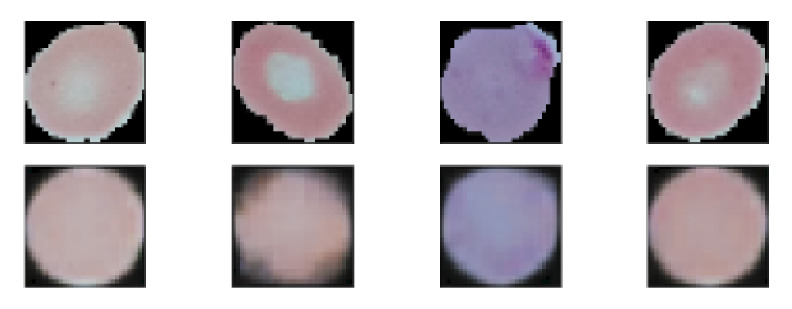
Some reconstructed images after training Autoencoder. Top images are the original image and bottom images are reconstructed images.

**Figure 10 diagnostics-10-00329-f010:**
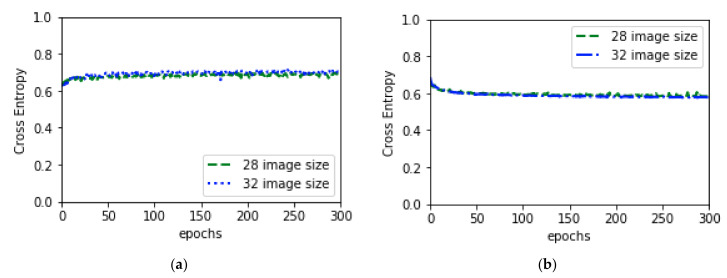
Training behavior of Autoencoder after freezing encoder layer: (**a**) depicts the validation accuracy for image (32, 32) and (28, 28) and (**b**) depicts validation loss for image (32, 32) and (28, 28) while training Autoencoder 300 epochs.

**Figure 11 diagnostics-10-00329-f011:**
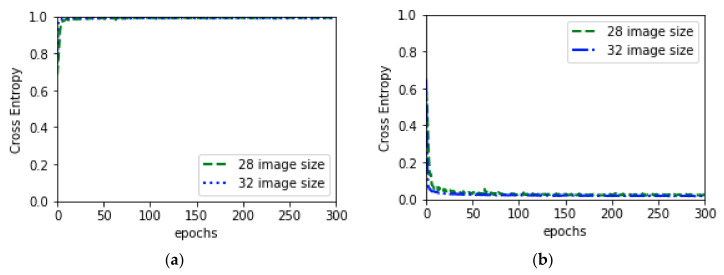
Final behavior of Autoencoder: (**a**) depicts the combine validation accuracy and (**b**) depicts combine validation loss while training Autoencoder 500 epochs.

**Figure 12 diagnostics-10-00329-f012:**
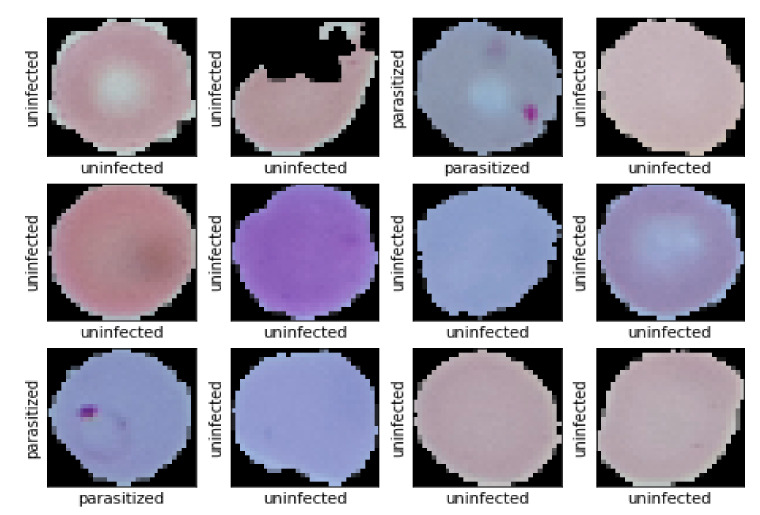
Some predicted result of Autoencoder based model. Here X-axis represents Ground Truth value and Y-axis represents model prediction.

**Figure 13 diagnostics-10-00329-f013:**
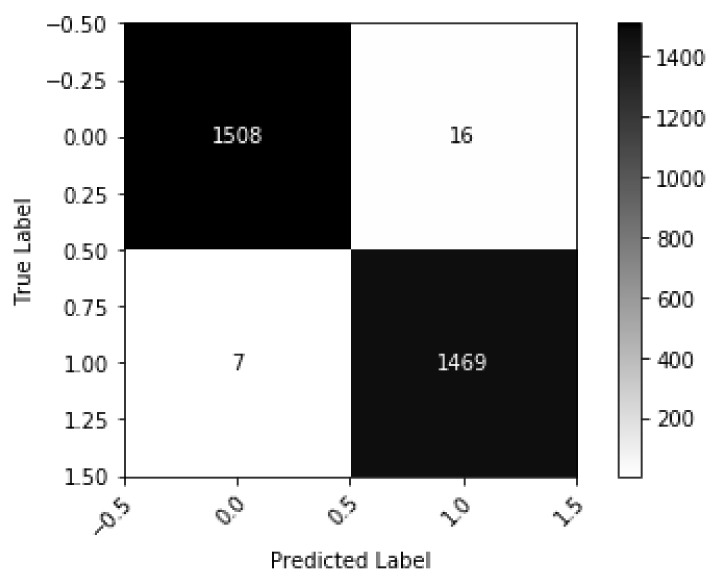
Confusion Matrix of Autoencoder based model based on test data.

**Figure 14 diagnostics-10-00329-f014:**
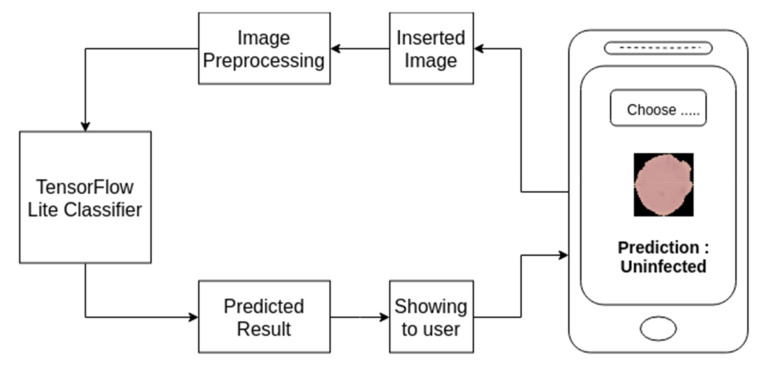
Outline of mobile based service.

**Figure 15 diagnostics-10-00329-f015:**
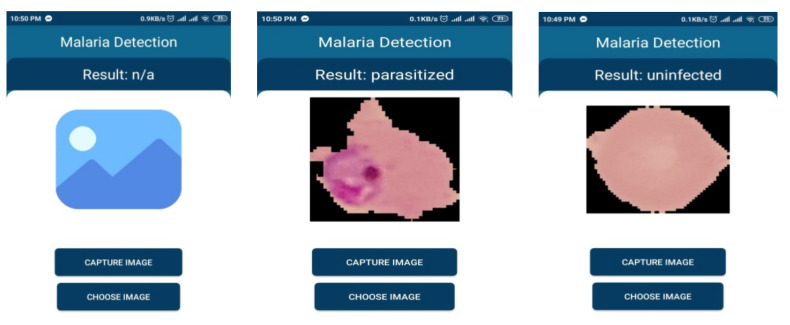
Model behavior on smartphone-based service.

**Figure 16 diagnostics-10-00329-f016:**
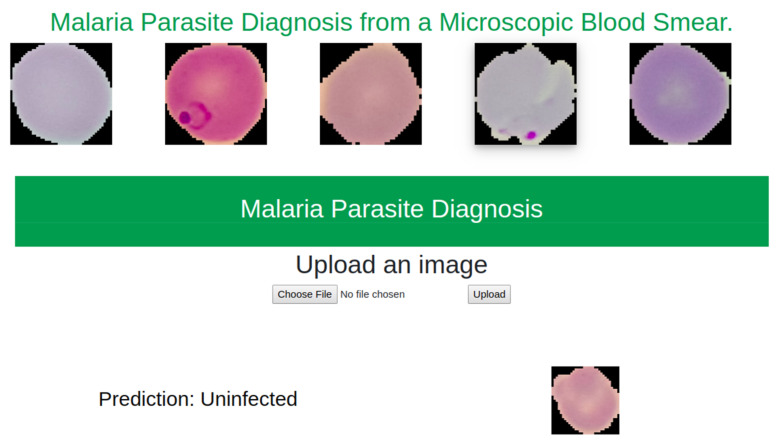
Model behavior on web based service.

**Table 1 diagnostics-10-00329-t001:** Augmentation table.

Augmentation Type	Parameters
Random Rotation	20 Degree
Random Zoom	0.05
Width Shift	(0.05, −0.05)
Height Shift	(0.05, −0.05)
Shear Intensity	0.05
Horizontal Flip	True

**Table 2 diagnostics-10-00329-t002:** Model performances after general training.

Image Size	Augmentation	Epochs	Validation Accuracy	Validation Loss
(32,32)	No	300	0.9982	0.06
(64,64)	No	300	0.9748	0.16
(32,32)	Yes	300	0.9920	0.02
(64,64)	Yes	300	0.9813	0.08

**Table 3 diagnostics-10-00329-t003:** Model performances after performing Knowledge Distillation.

Image Size	Augmentation	Epochs	Validation Accuracy	Validation Loss
(32,32)	Yes	500	0.9897	0.04
(64,64)	Yes	500	0.9859	0.05

**Table 4 diagnostics-10-00329-t004:** Model performances after training Autoencoder 500 epoch.

Image Size	Training Accuracy	Training Loss	Validation Accuracy	Validation Loss
(28,28)	0.9943	0.018	0.9924	0.025
(32,32)	0.9941	0.019	0.9900	0.032

**Table 5 diagnostics-10-00329-t005:** Model performances on different experiment.

Image Size	Aug	Method	Test Acc	Test Loss	F1 Score	Precs.	Sens.	Spec.	Size (KB)
(32,32)	Yes	-	0.9915	0.03	0.9914	0.9861	0.9960	0.9865	233.60
(32,32)	No	-	0.9877	0.05	0.9876	0.9892	0.9861	0.9893	233.60
(64,64)	Yes	-	0.9843	0.07	0.9839	0.9836	0.9840	0.9842	954.50
(64,64)	No	-	0.9755	0.15	0.9751	0.9851	0.9650	0.9855	954.60
(32,32)	Yes	Distillation	0.9900	0.04	0.9900	0.9877	0.9920	0.9878	233.60
(64,64)	Yes	Distillation	0.9885	0.04	0.9882	0.9929	0.9836	0.9932	954.60
(28,28)	Yes	Autoencoder	0.9950	0.01	0.9951	0.9929	0.9880	0.9917	73.70
(32,3)	Yes	Autoencoder	0.9923	0.02	0.9922	0.9892	0.9952	0.9917	73.70
(32,32)	Yes	CNN-SVM	0.9893	-	0.9918	0.9921	0.9916	-	-
(32,32)	Yes	CNN-KNN	0.9912	-	0.9928	0.9911	0.9923	-	-

**Table 6 diagnostics-10-00329-t006:** Compatibility of developed model in different devices.

Version	Codename	API	Distribution	Device	Model	Ram	Comments
2.3.3–2.3.7	Gingerbread	10	0.30%	-	-	-	Almost obsolete OS
4.0.3–4.0.4	Ice Cream Sandwich	15	0.30%	-	-	-	Almost obsolete OS
4.1.X	Jelly Bean	16	1.20%	-	-	-	Almost obsolete OS
4.4	Kitkat	19	6.90%	Samsung	GT-190601	2 GB	Work Perfectly
5	Lollipop	21	3.00%	Google	Nexus 4 (Emulator)	2 GB	Work Perfectly
6	Marshmallow	23	16.90%	Google	Nexus 5 (Emulator)	1 GB	Work Perfectly
7	Nougat	24	11.40%	Xiaomi	Readmi Note 4	3 GB	Work Perfectly
8	Oreo	26	12.90%	Samsung	S8 (Emulator)	2 GB	Work Perfectly
8.1	-	27	15.40%	LG	LM-X415L	2 GB	Work Perfectly but latency is higher compared to other devices.
9	Pie	28	10.40%	Xiaomi	A2 Lite	4 GB	Work Perfectly

**Table 7 diagnostics-10-00329-t007:** Performance comparison.

Method	Accuracy	Sensitivity	Specificity	F1 Score	Precision
Ross et al., (2006) [73]	0.730	0.850	-	-	-
Das et al., (2013) [74]	0.840	0.981	0.689	-	-
SS. Devi et al., (2016) [75]	0.9632	0.9287	0.9679	0.8531	-
Zhaohuiet al., (2016) [54]	0.9737	0.9699	0.9775	0.9736	0.9736
Bibin et al., (2017) [31]	0.963	0.976	0.959	-	-
Gopakumar et al., (2018) [58]	0.977	0.971	0.985	-	-
S. Rajaraman et al., (2019) [40]	0.995	0.971	0.985	-	-
Proposed Model	0.9923	0.9952	0.9917	0.9922	0.9892

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
