# Peer review of "Deep Learning Based Automatic Malaria Parasite Detection from Blood Smear and Its Smartphone Based Application"

_diagnostics, 2020, doi:10.3390/diagnostics10050329_

Round 1

Reviewer 1 Report

This manuscript aims to use convolutional neuronal network based methods to develop a method for diagnosis of malaria parasitemia from microscopy slides. The authors are particularly interested in producing an application that is suitable for settings where resources and settings are limited.

The authors develop a model that is excellent in its performance. It performs comparably to any of the previously published models that attempt to accomplish similar aims. The main concern for publication is that of novelty. The authors do attempt to highlight the novelty of the output in the paragraph starting on Line 77 but fail to do so convincingly. Part of this may be the grammar and diction but the statement that “[49] has dependency on captured microscopic images as their training images was collected under certain controlled condition” is not well explained. Are not the results of this submitted manuscript also “dependent on captured microscopy images”. And were not the training images here also “collected under certain controlled condition(s)”. A clear, concise explanation of how the methodology and output of this piece of work advances the scientific beyond what is already published in the area is needed.

Other smaller concerns:

Remove Bangladesh specific malaria data and multiple instances of mentioning how it would be useful in the ‘hills of Bangladesh’. If this is to be published in an international journal it should have international implications beyond Bangladesh.

In referencing papers, the format “Smith et al found” should be used rather than “[X] found”.

I would discourage abbreviations in the title.

The data set (referenced in 11) is several times referred to as “NIH malaria data”. It is unclear why that designation is used as it appears to have been collected in Thailand and stored in the UK. How is the NIH involved? And why is that nomenclature used?

Author Response

Thank you professor for your kind review. Attachment in bellow, we provided point-by-point response.

1) The authors do attempt to highlight the novelty of the output in the paragraph starting on Line 77 but fail to do so convincingly.
Thank you, professor for your review. We agree that we failed to highlight our novelty convincingly. We have reviewed and rewritten that part of the Introduction which are now available in line 75 - 82.

2) Grammar and diction problem
Thank you, Sir, for pointing out this. Considering your review, we have rechecked our script a few times and corrected grammatical mistakes.

3) Statement that “[49] has dependency on captured microscopic images as their training images was collected under certain controlled condition” is not well explained.“Dependent on captured microscopy images” is not well explained.“ Collected under certain controlled condition(s)” need a clear and concise explanation.
We also have realized that the mentioned statement is unclear; hence it seems irrelevant. We have paraphrased this statement and tried to state the differences between our work and cited paper. We also have compared the performances and techniques between these two works. All of these modifications can be found in line 63 - 71.

4) A clear, concise explanation of how the methodology and output of this piece of work advances the scientific beyond what is already published in the area is needed.
Thank you, Sir, for the valuable comment. We agree that we failed to provide a clear and concise explanation of this matter. Our motivation is to develop a complete end to end solution for automatic detection of the malaria parasite. In this regard, we developed a model which is compatible with deploying in a lower-cost platform, ex: a smartphone and able to perform offline. While developing a model, we also improved the performances compared with the existing solution. We also validated our model by testing on different platforms like different mobile version, web version etc. As per the best of our knowledge, none did that early. Previous works reported a high performing model without demonstrating how their model could perform in the resource-restricted area. On the other hand, one of our main focus was how the developed solution could easily fit into a low-cost platform. We have done so by an extensive experiment done on different platforms. (further discussed in the Discussion section). We reviewed our article and tried to highlight this. Explicitly it can be found in line 46 to 73, line 84 to 89 and line 444 to 468.

5) Remove Bangladesh specific malaria data and multiple instances of mentioning how it would be useful in the ‘hills of Bangladesh’. If this is to be published in an international journal it should have international implications beyond Bangladesh.
Thank you, professor, for your kind note. In the revised article, we changed the Bangladeshi perspective; instead, we focused on the unreached and less facilitated people of the developing nations. Bangladesh is one of the significant malaria-prone developing countries; therefore, we mentioned it few times to use as an example line number 40, 42 and 52.

6) In referencing papers, the format “Smith et al found” should be used rather than “[X]
found”.
Thank you, Sir. We considered your review and revised our article. In revised article, we used the suggested citation format, and this can be found in line 113, 115.

7) Abbreviations problem in the title.
Thank you Sir for your review. Based on your review, we changed our title from CNN based to Deep learning-based approach.

8) The data set (referenced in 11) is several times referred to as “NIH malaria data”. It is unclear why that designation is used as it appears to have been collected in Thailand and stored in the UK.
Thank you, professor, for pointing out this vital point. This publicly available dataset is hosted in NIH website, and it was publicly accessible by (Rajaramon et al. 2018). We collected this data from NIH website (link given below) hence several times we referred this data as NIH malaria data. Considering your review, we paraphrased this term, and instead of NIH malaria data, we mentioned this dataset as publicly available malaria data.
Link : https://lhncbc.nlm.nih.gov/publication/pub9932

Reviewer 2 Report

Generally, there is need for proper proof reading of the paper to use appropriate English language expression and tenses.  Some statements were repeatedly used. Abbreviations should be first defined before being used e.g. SVM in line 70.

Introduction

This is way too long and complex. This needs to be cut down to provide only the relevant information.  The concept of the work is not clearly demonstrated and hence makes it very ambiguous to comprehend. The significance of the work is lacking. The authors stated that the contribution of the work is expected to reduce malaria caused death. How so is this? They must really explain this. Lines 110-113 are very irrelevant.

Methodology

The authors should refine their description of the methods to make it comprehensible. Major English language revision is needed here as well. What do the authors mean by “Some instances of NIH data” in Figure 1?

Results

These would need better description

Discussion

This aspect of the article needs revision. It appears some aspects of the results were repeated in this section. The authors need to discuss their findings in line with the objectives and aims of their study which is clearly lacking. It must be clearly presented also in line with their results and needs to demonstrate its relevance and application.  

Author Response

1) There is need for proper proof reading of the paper to use appropriate English language expression and tenses.
Thank you, Sir, for your kind review. We agree that there were several grammatical mistakes and we are sorry for that. We have improved our script by correcting the errors.

2) Some statements were repeatedly used.
We have revised our full manuscript and removed repeated and unnecessary lines.

3) Abbreviations should be first defined before being used e.g. SVM in line 70.
Thank you, professor, for pointing out this. We have scanned the full manuscript and documented the abbreviations and its first appearances in the manuscript and written complete form at its first appearances.
CNN – First appear in Line 15 (Abstraction) SVM, KNN – First appear in Line 19 (Abstraction)
WHO – Line 36
RDT – Line 48
PCR – Line 48
SVM – Line 65 (Introduction)
PCA – Line 99
CNN – Line 105 (Introduction)
MSE – Line 189
KNN – Line 228 (Training Details)
SGD – Line 251

4) Introduction is way too long and complex. Introduction needs to be cut down to provide only the relevant information.
Thank you, professor, for your kind reply. We are sorry to hear this. To address this issue we have changed the overall structure of the Introduction. The improved version is now more specific and would give the readers a clear idea about the original problem, conventional way of addressing it and the significant contributions of this work. We added a separate literature review section to explore related works. This section will also help the readers understand the novelty of our contributions.

5) The concept of the work is not clearly demonstrated and hence makes it very ambiguous to comprehend.( Introduction) We agree with your statements. In Introduction, we could not make our audience clear about our aim and objectives. To overcome this, we have rewritten introduction section. Irrelevant and ambiguous sentences or information has been removed from Introduction.
6) The significance of the work is lacking in Introduction
Thank you, professor, for your review. We are sorry to learn that the significance of our contribution did not reflect convincingly. We have reviewed this and rewritten introduction section to highlight the novelty which is now available in line 75 to 82.

7) The authors stated that the contribution of the work is expected to reduce malaria caused death. How so is this? They must really explain this. (Introduction)
Sorry for the confusion. Our writing did not carry the proper information that we wanted to convey. Considering your review, we paraphrased this line. We wanted to send a message to the readers that our proposed system will help to diagnosis malaria parasite remotely and rapidly without requiring any expertise. So this will help underprivileged people to diagnosis quickly and adequately in resource-restricted areas. Since the diagnosis is rapid, the treatment could also be started quickly, which is not possible with the conventional approach. This is obvious that precautionary or proper treatment will reduce malaria caused death. We tried to highlight this message but failed to do convincingly. In the revised manuscript we have rewritten this line, and this can be found in line 84 – 89.

8) Lines 110-113 are very irrelevant.
Thank you Sir. Based on your review we have removed the respective lines

9) The authors should refine their description of the methods to make it comprehensible. (Methodology)
Thank you, Sir, for your review. Based on your review, we refined our methodology sections. New information (highlighted) have been added, which will be useful for the
readers to understand this section better. We also rewrite a couple of sentences to make it more compact.

10) Major English language revision is needed here as well. (Methodology) Thank you, Sir. We have rechecked our manuscript a couple of times and corrected grammatical problems.

11) What do the authors mean by “Some instances of NIH data” in Figure 1? (Methodology)
Sorry for the confusion. It means the representative images from the original dataset.

12) These would need better description (Results) Thank you Sir for your kind review. We also believe that a better description could be given in the result section. Considering your review, we revised our result section and improved its description.

13) Discussion needs revision. In Discussion, It appears some aspects of the results were repeated. The authors need to discuss their findings in line with the objectives and aims of their study which is clearly lacking. (in Discussion)
Thank you, professor, for the essential note. We agree with you that some of the lines were used repeatedly in this section. Besides that, we also realize that we did not discuss here our findings properly and failed to relate them with the aims of this work. We have rewritten this section and tried to discuss our results in line with aims and objectives. We also have highlighted our success in terms of performances and model effectiveness. Rewritten discussion can be found in line 444 – 468.

14) Findings and Objectives must be clearly presented also in line with their results and needs to demonstrate its relevance and application.
We completely agree with your statement. Earlier our findings and objectives were not discussed properly. Therefore, we have rewritten Introduction and discussion section to present our objectives and findings clearly. Our changes can be found in line 75-82 and line 444 – 468.

Round 2

Reviewer 1 Report

Fuhad Diagnostics revision

This is the revision of a manuscript previously submitted to the Journal.  Although it has improved significantly from the first round – I feel that there remain significant weaknesses preventing it from being published in its current form.

The utility of the diagnostic needs to be better explained. As I understand it – this software is VERY good at telling me if a single cell is infected or uninfected. This is useful. Perhaps I have missed some of the details though. As a clinician I do not want to know if a single cell is infected. I want to know if ANY of the cells in a smear are infected. The details of determining this are somewhat different from the details of determining if a single cell is infected. Determining patient infection status sometimes relies on searching through 100 different high power microscope fields for a single infected cell. How does this software deal with that? Do 100 different images need to be uploaded?

Determination of infection status using this method would still rely on the preparation of a high quality microscopic slide – which is the hardest part of visual detection of malaria parasites. The technique proposed is not low tech – it would still require human resources and expertise to make and stain a slide (requires glass slides, fingerprickers, expertise at smear preparation, stain, a high quality 100X microscope, and electricity). I am not sure that this expertise is going to be available in the rural hill areas where the authors are proposing the use of this algorithm. It is also still unclear to me how the images are going to be loaded into the cell phone. Is the cell phone put to the eyepiece of the microscope? An additional complication is that many fields will have cells that are touching each other. The ideal RBC untouched by another red cell, as shown in all of the figures, is the exception rather than the rule. Is the image processing software able to deal with ‘confluent’ cells?

If the authors continue to want to proclaim that this technique is less high tech and more applicable to rural areas than other diagnostic techniques – they need to compare it to the RDT. The RDT requires no extra equipment (other than a fingerpricker), has results in 10 minutes, needs minimal training, and can be performed without electricity. I am afraid that I do not see the advantage of this technique over the RDT. I am afraid that I really do not see the ability of this algorithm to move diagnosis into a more rural setting. It still relies on high quality microscopic images, which are very difficult to obtain.

There remain English grammar and diction issues. Although the manuscript reads much better than the first edition, it still needs a thorough editing. Some examples listed below:

Line 12 “electron microscopy”? I am not sure that is the term you are looking for – electron microscopes are very rarely used in malaria diagnostics

The term “etc” should not be used in the abstract. The authors should list all of the techniques that they used or merely state “a variety of techniques”

Line 20 and 25 – not ‘proposed’ is it? At this point you have developed the model. State “our model” or “the model”

Line 35 – not “referred to as” they ARE the malaria vector

Line 49 – “not compatible with modern day’s techniques” is a confusing phrase

Line 59 – what reported solution are you referring to?

Line 85 “very early stage”? Again – as listed above I am unclear as to how this model will move diagnosis any further into the rural areas as it still relies on electricity, significant human resources and a high-quality microscope.

Author Response

Thank you professor for your kind review. According to your review we provide a point by point response.

  1. The utility of the diagnostic needs to be better explained. As I understand it – this software is VERY good at telling me if a single cell is infected or uninfected. This is useful. Perhaps I have missed some of the details though. As a clinician I do not want to know if a single cell is infected. I want to know if ANY of the cells in a smear are infected. The details of determining this are somewhat different from the details of determining if a single cell is infected. Determining patient infection status sometimes relies on searching through 100 different high power microscope fields for a single infected cell. How does this software deal with that? Do 100 different images need to be uploaded?

Response: Thank you professor for your kind review. We agree that this has not been clarified appropriately. The concentration of this work was to develop an efficient and highly accurate model which can detect malaria parasites from a single segmented cell image. We have demonstrated that it is possible to do so with high accuracy and low computational cost (4600 flops as opposed to 19.6 billion flops). A complete system that also includes image acquisition and cell segmentation is out of the scope of this work. But we believe, the contribution made in this work will be valuable in eventually realizing such a complete system which will be efficient and cost effective. The paper has been modified to better reflect this in lines 53-65, 471-476.

  1. Determination of infection status using this method would still rely on the preparation of a high quality microscopic slide – which is the hardest part of visual detection of malaria parasites. The technique proposed is not low tech – it would still require human resources and expertise to make and stain a slide (requires glass slides, fingerprickers, expertise at smear preparation, stain, a high quality 100X microscope, and electricity). I am not sure that this expertise is going to be available in the rural hill areas where the authors are proposing the use of this algorithm. It is also still unclear to me how the images are going to be loaded into the cell phone. Is the cell phone put to the eyepiece of the microscope? An additional complication is that many fields will have cells that are touching each other. The ideal RBC untouched by another red cell, as shown in all of the figures, is the exception rather than the rule. Is the image processing software able to deal with ‘confluent’ cells?

Response: Thank you professor for your kind review. We agree that it was not been clarified properly. Yes, we envision that in such a system the cell phone is placed in the eyepiece of Microscope for image acquisition purpose as shown in [1, 2]. This has been now been clarified in the paper with reference in line 138 and 53. As for the cell segmentation issue, it has been addressed by numerous other reports and is not scope of our current work. In near future, with a help of portable acquisition system and segmentation algorithm we will be able to develop a complete automated system for malaria parasite detection. We have now clarified it in line 79-80, 116-120, 474-476 in the paper.

  1. If the authors continue to want to proclaim that this technique is less high tech and more applicable to rural areas than other diagnostic techniques – they need to compare it to the RDT. The RDT requires no extra equipment (other than a fingerpricker), has results in 10 minutes, needs minimal training, and can be performed without electricity. I am afraid that I do not see the advantage of this technique over the RDT. I am afraid that I really do not see the ability of this algorithm to move diagnosis into a more rural setting. It still relies on high quality microscopic images, which are very difficult to obtain.

Response: Thank you professor for your kind review. Considering your review we also think that we should address and compare RDT with our solutions. In revised paper in line 45-53 we have addressed this issue. Also, in line with your recommendation we have now reduced the mention of rural area.

  1. Line 12 “electron microscopy”? I am not sure that is the term you are looking for – electron microscopes are very rarely used in malaria diagnostics

Response: Thank you professor for pointing this. We agree that, it was incorrectly referred to “electron microscope” in the older version which has now been corrected and can be found in line 12, 47.

  1. The term “etc” should not be used in the abstract. The authors should list all of the techniques that they used or merely state “a variety of techniques”

Response: Thank you professor. In revised manuscript we rewrite abstract section where this problem has been resolved.

  1. Line 20 and 25 – not ‘proposed’ is it? At this point you have developed the model. State “our model” or “the model”

Response: Thank you professor. According to your review we removed this term and revised full abstract section.

  1. Line 35 – not “referred to as” they ARE the malaria vector

Response: Thank you professor for your kind review. We also think that we should not use the term referred as. Hence not only paraphrased this term but also we revised full introduction part.

  1. Line 49 – “not compatible with modern day’s techniques” is a confusing phrase

Response: Thank you professor for your kind review. We also think that this phrase is confusing and we failed to clarify this confusion. Therefore, in new version we corrected and rephrased this term.

  1. Line 59 – what reported solution are you referring to?

Response: Thank you professor for you kind review. In our early version, we stated reported solution where reported solution referred reported automatic malaria parasite detection solution. In revised manuscript we rewrite introduction part and solved this issue.

  1. Line 85 “very early stage”? Again – as listed above I am unclear as to how this model will move diagnosis any further into the rural areas as it still relies on electricity, significant human resources and a high-quality microscope.

Response: Thank you professor for your kind review. Based on the review we think your stated line seems confusing and unclear. Hence in this revised version we removed those lines to present our work more clearly to the readers.

References

  1. Quinn, J.A., Nakasi, R., Mugagga, P.K., Byanyima, P., Lubega, W., Andama, A.: Deep convolutional neural networks for microscopy-based point of care diagnostics. In: Machine Learning for Healthcare Conference.2016, pp. 271–281.
  2. Rosado, L., Da Costa, J.M.C., Elias, D., Cardoso, J.S.: Automated detection of malaria parasites on thick blood smears via mobile devices. Procedia Computer Science 90, 2016, 138–144

Reviewer 2 Report

Abstract

 Line 28 Change Malaria to malaria and correct “in the resource” to “resource restricted”

Introduction

Rephrase line 48

Line 70 appears incomplete

Check line 86

Check the statement at 105-106

Why the 2 different parts. Can it not be merged and cut down? The beginning of the second part still sounded more like a repetition.

Methodology

Why the use of some instances?

Line 263 playes

Results and discussion

Makes some sense now but still needs some text revisions especially in the discussion part.

Author Response

Thank you professor for your kind review. According to your review we provide a point by point response.

1. Line 28 Change Malaria to malaria and correct “in the resource” to “resource restricted” (Abstract)

Response: Thank you professor. We revised the complete abstract section and resolved this issue.

2. Rephrase line 48 (Introduction)

Response: Thank you professor. Considering your review we revised full introduction section and resolved this issue.

3. Line 70 appears incomplete (Introduction)

Response: Thank you professor. It was an unintentional mistake. In new version, we revised introduction section and corrected the errors.

4. Check line 86 (Introduction)

Response: Thank you professor for your kind review. Based on the review we think that line 84-89 seems confusing and unclear. Hence in this revised version we removed those lines to present our work more clearly to the readers.

5. Check the statement at 105-106. Why the 2 different parts. Can it not be merged and cut down? The beginning of the second part still sounded more like a repetition. (Introduction)

Response: Thank you professor. We also agree that it sounded like repetition. We merged these two lines in the new version and solved this issue.

6. Why the use of some instances? (Methodology)

Response: Thank you professor. We removed this term and resolved this issue.

7. Line 263 playes

Response: Thank you professor. It has been corrected.

8. Makes some sense now but still needs some text revisions especially in the discussion part. (Results and Discussion)

Response: Thank you professor. As per your recommendation, we revised multiple section including abstract, introduction, related work, discussion and conclusion.
